# A Meta-Analysis of the Reliability of Four Field-Based Trunk Extension Endurance Tests

**DOI:** 10.3390/ijerph17093088

**Published:** 2020-04-29

**Authors:** María Teresa Martínez-Romero, Francisco Ayala, Mark De Ste Croix, Francisco J. Vera-Garcia, Pilar Sainz de Baranda, Fernando Santonja-Medina, Julio Sánchez-Meca

**Affiliations:** 1Department of Physical Activity and Sport, Faculty of Sports Sciences, University of Murcia, 30720 San Javier (Murcia), Spain; 2Department of Sport Science, Sports Research Centre, Miguel Hernández University of Elche, 03202 Elche (Alicante), Spain; 3School of Sport and Exercise, Exercise and Sport Research Centre, University of Gloucestershire, Gloucester GL2 9HW, UK; 4Department of Surgery, Pediatrics, Obstetrics and Gynecology, Faculty of Medicine, University of Murcia, 30100 Murcia, Spain; 5Traumatology and Orthopedic Surgery Service, Virgen de la Arrixaca University Clinical Hospital, 30120 Murcia, Spain; 6Department of Basic Psychology and Methodology, Faculty of Psychology, University of Murcia, 30100 Murcia, Spain

**Keywords:** reliability generalization meta-analysis, core endurance, physical education, sports performance, pre-participation assessment

## Abstract

This meta-analysis aimed to estimate the inter- and intra-tester reliability of endurance measures obtained through trunk extension field-based tests and to explore the influence of the moderators on the reliability estimates. The reliability induction rate of trunk extension endurance measures was also calculated. A systematic search was conducted using various databases, and subsequently 28 studies were selected that reported intraclass correlation coefficients for trunk extension endurance measures. Separate meta-analyses were conducted using a random-effects model. When possible, analyses of potential moderator variables were carried out. The inter-tester average reliability of the endurance measure obtained from the Biering-Sorensen test was intraclass correlation coefficient (ICC) = 0.94. The intra-session reliability estimates of the endurance measures recorded using the Biering-Sorensen test, the prone isometric chest raise test, and the prone double straight-leg test were ICC = 0.88, 0.90, and 0.86, respectively. The inter-session average reliability of the endurance measures from the Biering-Sorensen test, the prone isometric chest raise test, and the dynamic extensor endurance test were ICC = 0.88, 0.95, and 0.99, respectively. However, due to the limited evidence available, the reliability estimates of the measures obtained through the prone isometric chest raise, prone double straight-leg, and dynamic extensor endurance tests should be considered with a degree of caution. Position control instruments, tools, and familiarization session demonstrated a statistical association with the inter-session reliability of the Biering-Sorensen test. The reliability induction rate was 72.8%. Only the trunk extension endurance measure obtained through the Biering-Sorensen test presented sufficient scientific evidence in terms of reliability to justify its use for research and practical purposes.

## 1. Introduction

In the last two decades, there has been an increased interest in the assessment of trunk extensor muscle endurance, as deficits in trunk extensor endurance and imbalances between trunk muscle groups have been suggested as being primary risk factors for low back disorders [1,2,3,4] and could negatively affect sports performance [5,6].

Although several sophisticated laboratory-based tests have been developed to quantify trunk extensor muscle endurance (i.e., force platforms, isokinetic dynamometers) [7,8,9], field-based tests seem to be the most popular tests, probably because they are portable, cost-effective, easy to use, and time-efficient methods [10]. Different trunk extension endurance field-based tests have been widely used, including (a) isometric endurance tests (i.e., Biering-Sorensen test [2], prone isometric chest raise test [11], and prone double straight-leg raise test [12]), which involve maintaining a position against gravity for as long as possible, and (b) dynamic endurance tests (i.e., dynamic extensor endurance test [3,13]), which consist of performing as many repetitions as possible in a given time or with a certain cadence until exhaustion. However, the validity and reliability of their outcomes must be determined [14] before these field-based tests can be used to identify deficits in trunk extensor muscle endurance, in order to explore imbalances between trunk muscle groups and to establish progress from training and/or rehabilitation programs. The four trunk extension endurance measures identified have been considered operationally valid by medical (American College of Sports Medicine [15]), sport (Swiss Olympic Medical Centers [16]), and educational (Cooper Institute [17]) organizations. Furthermore, their measures have been shown to be sensitive enough to detect trunk extensor muscle endurance deficits in patients with chronic low back pain [18,19], and are included in many prominent sports medicine textbooks [15,16,17].

Two different types of reliability should be considered before choosing an appropriate test for research and clinical purposes [20,21]: inter-tester reliability and intra-tester reliability. Inter-tester reliability provides information regarding the degree to which measures taken by different testers using identical test protocols on the same cohort of individuals are similar or consistent [20]. Intra-tester reliability provides information regarding the degree to which several measurements taken at different times for the same test by the same tester are similar [22]. The intra-tester reliability can be determined using short (generally within a day: intra-session) or long (generally more than one day: inter-session or test–retest) time intervals to separate the testing sessions [14,20].

In the scientific literature, several studies have examined the inter- [18,23,24] and intra-tester (intra- [18,25,26] and inter-session [9,11,27]) reliability of the four field-based trunk extension endurance tests in different cohorts of individuals (e.g., athletes, children, adolescents, sedentary adults). The findings of these studies, and for both types of reliability, have shown a large degree of heterogeneity, with intraclass correlation coefficient (ICC) scores ranging from 0.77 to 0.99 and from 0.20 to 0.99 for the inter- and intra-tester reliability, respectively.

The large degree of heterogeneity presented in the ICC scores reported suggests that the reliability outcomes of a physical performance measure could be influenced by several factors (the tester’s experience or training in administering the test, variations in the assessment methodology, and participant-related variability, etc.). If practitioners do not consider these factors, they might select a trunk extension endurance test with an inter- and intra-tester reliability that is inappropriate for their given population. In practical terms, the selection of an inappropriate test can lead not only to an inaccurate diagnosis of normality or deficits but also an incorrect evaluation of the effectiveness of a training or rehabilitation intervention for improving or maintaining trunk extensor endurance. Frequently, studies exploring trunk extension endurance do not report the reliability estimates of the test they are using with the given population [28,29,30], or the reliability coefficients obtained in previous studies are simply cited [3,31,32]. This characteristic of referencing the reliability coefficients from prior studies instead of reporting the reliability obtained with their data has been coined reliability induction (RI) [33].

The reliability generalization (RG) is a meta-analytical approach that emerges as a criticism of the widespread practice of RI. The purpose of this method is to estimate the average reliability of the scores of a given test, as well as to determine the variability of the reliability coefficients reported by the different studies that have used this test. Moreover, if the variability is very high, another aim is to explore which characteristics of the studies may be statistically associated with the reliability estimates [34,35,36]. To the best of the authors’ knowledge, there are no RG meta-analyses published to date concerning trunk extension endurance measures obtained through field-based tests. This information might be useful for practitioners because it can aid them in deciding on the best field-based test/s (in term of reliability) to assess trunk extensor muscle endurance according to the characteristics of their patients or athletes.

Therefore, the main purpose of the current study was to conduct a RG meta-analysis (a) to obtain combined reliability estimates of trunk extension endurance measures obtained through four field-based tests, (b) to identify which characteristics of the studies may influence the variability of the reliability coefficients, and (c) to determine the RI practice in studies that have used trunk extensor muscle endurance tests.

## 2. Materials and Methods

The current RG meta-analysis (PROSPERO ID: CRD42019123179) was conducted following the Preferred Reporting Items for Systematic Reviews and Meta-Analyses (PRISMA) guidelines [37]. The PRISMA checklist is presented in Appendix A.

### 2.1. Study Selection

In this RG meta-analysis, the following inclusion criteria were considered: (a) being an empirical research study (psychometric study) in which the original and modified versions of the following field-based tests were applied: Biering-Sorensen test [2], prone isometric chest raise test [11], prone double straight-leg raise test [12], and dynamic extensor endurance test [3] (a detailed description of each test is presented in Box 1); (b) being written in English or Spanish; (c) being published before March 2019; (d) using a sample of at least 10 participants; and (e) reporting the ICC as a measure of reliability (for both the inter- and intra-tester (intra-session or internal consistency and inter-session or test-retest reliability) reliability) of any of the aforementioned field-based tests or providing sufficient data from which this coefficient could be calculated through standardized equations. The selection of the ICC as a valid measure of reliability was made following the consensus-based standards for the selection of health measurement instruments (COSMIN) statement [38,39,40]. Studies whose main objective was not the analysis of the reliability of the trunk extension endurance measures obtained through any of the four aforementioned field-based tests, but which reported reliability data were also eligible for inclusion. The same inclusion criteria were considered for selecting studies that induced reliability, except (d) and (e).

Literature reviews, abstracts, editorial commentaries, and letters to the editor were excluded. Finally, some authors were contacted to provide missing data or to clarify if data were duplicated in other publications. Incomplete data or data from an already included study were also excluded.

### 2.2. Search Strategy

A systematic computerized search was conducted up to 28 February 2019 in the databases MEDLINE, PubMed, Web of Science, and Scopus. In addition, a supplementary search in Google Scholar was also performed. Relevant keywords were used to construct Boolean search strategies, including terms such as reliability, extensor muscles, trunk, core, endurance, performance, and test (see Appendix A). Then, the reference lists of included studies were screened for potentially eligible studies. Two reviewers independently (M.T.M.-R. and P.S.d.B.) selected studies for inclusion in a two-step process. First, studies were screened on the basis of title and abstract. In a second stage, full-text studies were reviewed to identify those studies that met the eligibility criteria. Disagreements were resolved through consensus or by consulting a third reviewer (F.A.).

Box 1.Trunk extension endurance field-based tests description.

**Isometric Endurance Field-Based Tests**
Biering-Sorensen test

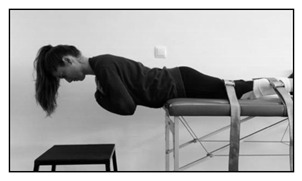

The test consists of assessing how many seconds the participant can keep the unsupported upper part of the body (from the upper border of the iliac crest) horizontal while placed prone with the buttocks and legs fixed to the table bench by three wide canvas straps, with the arms across the chest. The test is continued until the participant could no longer control his/her posture for a maximum of 240 s.Prone isometric chest raise test

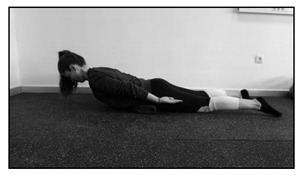

The test consists of assessing how many seconds the participant can keep the sternum off the floor while placed prone with the arms along the body. A small pillow is placed under the iliac crest to decrease the lumbar lordosis. The subject is asked to maintain the position for as long as possible, not exceeding a 5 min time limit.Prone double straight-leg raise test

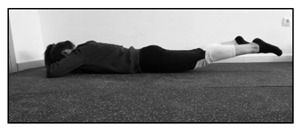

The test consists of assessing how many seconds the participant can keep both legs raised with the knees off the mat while placed prone with hips extended, the hands underneath the forehead and the arms perpendicular to the body. The test is continued until the participant can no longer maintain knee clearance. 
**Dynamic Endurance Field-Based Test**
Dynamic extensor endurance test

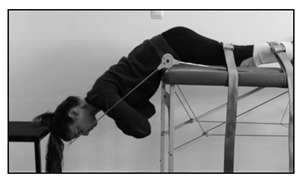

The test consists of assessing how many repetitions the participant can perform while placed prone with the unsupported upper part of the body (from the upper border of the iliac crest). The arms are positioned along the body and the buttocks and legs are fixed by three straps. With the spine kept straight, the subject is instructed to extend the trunk to neutral and then to lower the upper body 45 degrees. A repeated beat guided the subject to maintain a cadence of 25 repetitions per minute until exhaustion.


### 2.3. Data Extraction

To guarantee the maximum possible objectivity, a codebook was produced that specified the standards followed in coding each of the characteristics of the studies selected (see Appendix A). The moderator variables of the eligible studies were coded and grouped into four categories: (1) general study descriptors (authors, publication year, country, and study objective), (2) description of the study population (sample size, age, sex distribution, and target population), (3) description of the field-based test (version, protocol, and scores), and (4) type of reliability analyzed (inter-tester and/or intra-tester) and main characteristics of the study design (familiarization sessions, number of measurements, number of testers, time interval between measurements, test conditions, etc.).

### 2.4. Reliability Estimates

In this RG study, the ICC was extracted from the eligible studies as the only valid measure of inter- and intra-tester reliability. There are other coefficients such as the standard error of measure (SEM), Pearson correlation, and minimal detectable change with a 95% confidence level (MDC_95_) that have also been proposed as measures of reliability [20,41]. However, the scarce number of references that reported these types of reliability coefficients [42,43,44] did not allow for any separate meta-analyses to be conducted.

In order to maximize the number of studies included in this RG, when an article only reported the SEM, ICC was estimated by
(1)$$ICC=\sqrt{1-\frac{SEM^{2}}{SD^{2}}},$$
with *SD* being the standard deviation of the test scores.

Likewise, when an article only reported the MDC, it was transformed into SEM using the following formula:
(2)$$SEM=\frac{MDC}{1.96\times \sqrt{2}}$$

Later, the SEM was converted into ICC using the previously stated formula.

### 2.5. Quality Assessment

The COSMIN methodology was used to evaluate the quality of eligible studies for the RG meta-analysis, which consists of three sub-steps [38,39,40]. First, the risk of bias in each study was assessed using the COSMIN risk of bias check-list [39]. The check-list contains standards referring to design requirements and preferred statistical methods of studies on measurement properties. For each measurement property, a COSMIN box was developed, containing all standards needed to assess the quality of a study on that specific measurement property. Specifically, Appendix A regarding reliability was used for this RG meta-analysis. Each standard of the box was rated as very good, adequate, doubtful, or inadequate quality. To determine the overall rating of the quality of every single study, the lowest rating of any standard in the box was taken (i.e., “the worst score counts” principle) [45]. Second, from the data extracted on the description of the study population and the results on the reliability coefficients, the result per study was rated against the criteria for good measurement properties [38] as sufficient (+(ICC ≥ 0.70)), insufficient (−(ICC < 0.70)), or indeterminate (? (ICC not reported)). Finally, the results from different studies on one measurement property were statistically pooled in a meta-analysis and the quality of the evidence was graded (high, moderate, low, or very low evidence) using the modified Grading of Recommendations Assessment, Development and Evaluation (GRADE) approach. Four of the five GRADE factors were used in this meta-analysis: risk of bias (i.e., the methodological quality of the studies), inconsistency (i.e., unexplained inconsistency of results across studies), imprecision (i.e., total sample size of the available studies), and indirectness (i.e., evidence from different populations than the population of interest in the review). The fifth factor, publication bias, is difficult to assess in studies on measurement properties due to a lack of registries for these types of studies. Therefore, we did not take this factor into account in this meta-analysis [38,39,40]. The starting point is always the assumption that the pooled or overall result is of high quality. The quality of evidence is subsequently downgraded by one or two levels per factor to moderate, low, or very low when there is a risk of bias, inconsistency, imprecision, or indirect results [38]. Appendix A displays a brief description of each step of the COSMIN methodology.

The data extraction and quality assessment were double-coded (M.T.M.-R. and P.S.d.B.) to assess the inter-coder reliability of the coding process. Two authors working independently randomly coded 50% of the studies. For the quantitative moderator variables, ICC_3,1_ were calculated, whereas for the qualitative moderator variables, we applied Cohen’s kappa coefficients. In general, the agreement coefficients between the two authors were satisfactory as proposed by Orwin and Vevea [46], with the kappa coefficients ranging between 0.74 and 1, and the ICC ranging between 0.96 and 1. Inconsistencies between the two coders were resolved by consensus, and when these were due to ambiguity in the coding book, this was corrected. As before, any disagreement was resolved by mutual consent or in consultation with a third reviewer (F.A.).

### 2.6. Data Synthesis and Analysis

Separate meta-analyses were carried out for the different field-based tests and types of reliability analysis to avoid dependence problems, given that each study could report more than one reliability coefficient.

Statistical analyses were completed by assuming a random-effects model, and each reliability coefficient was weighted by the inverse variance [47]. Reliability coefficients were transformed into Fisher’s Z to normalize their distribution and to stabilize the variances [36]. For each meta-analysis, an average reliability coefficient (ICC+) and a 95% confidence interval (95% CI) were calculated [48].

The heterogeneity exhibited by the reliability coefficient, which represents the percentage of total variation across all studies due to between-study heterogeneity, was assessed by constructing a forest plot and by calculating the *Q* statistic and the *I^2^* index. Analyses of potential moderator variables were carried out when the *Q* statistic was statistically significant, the *I^2^* index was over 25%, and there were at least 20 reliability coefficients [49]. The only test that fulfilled these conditions was the Biering-Sorensen test for intra-tester reliability (k = 28 studies).

The influence of qualitative moderator variables (i.e., sample type (children and adolescents, adults), target population (community, clinical), physical activity level (sedentary, active), test duration (until exhaustion, until 240 s), among others) on the reliability coefficients were explored through analysis of variance (ANOVAs) and assuming mixed-effects model. Meta-regressions were also applied to test the influence of continuous moderators on the reliability coefficients assuming mixed-effects model, such as final sample size, average percentage of females, average age, percentage of attrition, number of measurements, number of testers, and time interval between measurements. *Q_B_* and *Q_w_* for ANOVAs and *Q_R_* and *Q_E_* statistics for meta-regressions were calculated to test the statistical significance of each moderator variable and to assess the model misspecification, respectively [50]. In addition, the proportion of variance accounted for by the moderator variables was estimated with *R^2^* following Raudenbush’s proposal [51]. In order to find the subset of moderator variables that can explain most of the reliability coefficient variability, a multiple meta-regression model (by assuming a mixed-effects model) was adjusted.

To facilitate the interpretation of the results, the average reliability coefficients and their confidence limits were back-transformed to the original metric of reliability coefficients. Furthermore, it has been suggested that for studies conducted in the sports medicine field of knowledge and aimed at analyzing the reliability (inter- and intra-tester) of quantitative physical performance measures, ICC values of 0.8 to 0.9 may be considered as acceptable, but values higher than 0.9 are desirable [14].

The statistical analyses were carried out with the software Comprehensive Meta-analysis 3.3 (BioStat, Englewood, NJ, USA) [52].

## 3. Results

### 3.1. Study Selection

A total of 1452 references were identified with all search strategies, from which 494 were excluded in the first screening as duplicates (34%). A total of 674 studies (46.4%) were eliminated after reading the title and abstract. Another 181 studies (12.4%) were removed after reading the full-text—120 studies did not use any trunk extensor muscle endurance field-based test (8.2%) and 61 articles did not meet the established inclusion criteria (4.2%) [53,54,55,56,57,58,59,60,61,62,63,64,65,66,67,68,69].

This search process identified 103 empirical studies that met the inclusion criteria (7.1%), in which 28 articles (1.9%) reported some reliability coefficients [8,9,10,11,18,23,24,25,26,27,43,53,54,55,56,57,58,59,60,61,62,63,64,65,66,67,68,69] (resulting in 43 cohort groups, as 9 studies had more than 1 group) and 75 articles (5.1%) induced the reliability (see Appendix A). Figure 1 shows the flow chart of the selection process of the studies.

### 3.2. Descriptive Characteristics of the Selected Studies for the RG Meta-Analysis

Appendix A provides a descriptive summary of the characteristics of the included studies for the RG meta-analysis. The studies selected were carried out between 1986 and 2018 and comprised participants from four continents (Europe [8,10,24,43,53,54,55,56,57,59,60,61,63,65,66], America [9,25,26,27,64,67,68,69], Asia [11,18,58,62], and Oceania [23]). Only one study was written in Spanish [60], and the rest of the selected studies were written in English [8,9,10,11,18,23,24,25,26,27,43,53,54,55,56,57,58,59,61,62,63,64,65,66,67,68,69].

The total sample size was 1097 participants, with an average of 25.5 subjects per cohort group (minimum = 10 [43,53,59,62,63] and maximum = 100 [11]). Three studies used healthy children [57] and adolescents [27,62]. Furthermore, 14 studies employed asymptomatic samples [9,10,18,24,25,27,53,56,57,59,60,62,63,68,69], 8 used patients with low back pain (LBP) [8,54,55,58,64,65,66,67], and 6 included both types of samples [11,23,26,43,61].

Concerning the type of field-based test used, 25 studies (89.2%) investigated the reliability of the back extensor endurance measures obtained through different modified versions of the Biering-Sorensen test [8,9,10,18,23,24,25,26,27,43,53,54,56,57,59,60,61,62,63,64,65,66,67,68,69], 4 studies (14.3%) used the prone isometric chest raise test (two of them used the original version [11,58] and another two used a modified version [18,55]), 2 studies employed a modified version of the dynamic extensor endurance test [27,62], (7.1%) and 1 study used the original prone double straight-leg raise test [18] (3.6%).

The most common types of reliability coefficients reported in the selected articles were the intra-tester and inter-session reliability (24 studies [8,9,10,11,24,26,27,43,53,54,55,56,57,58,59,60,61,62,63,64,65,66,68,69] that reported 37 coefficients), followed by the intra-tester and intra-session reliability (5 studies [18,25,26,43,67] that reported 16 coefficients), and the inter-tester reliability with 10 coefficients reported from 5 different studies [18,23,24,26,66].

### 3.3. Quality of the Selected Studies for the RG Meta-Analysis

Regarding the use of the Biering-Sorensen test in the studies, the COSMIN risk of bias checklist reported very good methodological quality scores for seven studies (five studies that evaluated the intra-tester and the inter-session reliability [8,10,59,61,66] and two studies for the inter-tester reliability [24,66]) and adequate quality scores in one study that analyzed the intra-tester and inter-session reliability [26]. In contrast, 13 studies [9,24,27,43,53,54,56,57,60,64,65,68,69] that analyzed the intra-tester and inter-session reliability obtained a doubtful methodological quality, as well as all of the studies that assessed the intra-tester and intra-session reliability [18,25,26,43,67] and three studies that evaluated the inter-tester reliability [18,23,26]. Two studies that analyzed the intra-tester and inter-session reliability showed inadequate methodological quality [62,63].

Concerning the use of the prone isometric chest raise test, three studies presented doubtful methodological quality scores for intra-tester and inter-session reliability [11,55,58], the same as for intra-tester and intra-session and inter-tester reliability where only one study was found [18]. It was the same with the prone double straight-leg test, which reported a doubtful quality for the only study found [18]. Finally, the two studies that used the dynamic extensor endurance test exhibited doubtful [27] and inadequate [62] methodological qualities for the intra-tester and inter-session reliability.

The detailed data of the COSMIN risk of bias check-list and criteria for good measurement properties are presented in Appendix A. Likewise, a summary of findings (SoF) per type of reliability and field-based test, including the pooled results of the measurement properties, the overall rating based on the inconsistency of results (i.e., sufficient (+), insufficient (−), or indeterminate (?)), and the grading of the quality of evidence (i.e., high, moderate, low, very low) is presented in Table 1. In general, the Biering-Sorensen test is the only test that presented moderate quality of evidence for inter-tester and intra-tester (inter-session) reliability.

### 3.4. Effect Sizes

#### 3.4.1. Primary Outcomes

Table 2 presents the average reliability of the trunk extensor measures obtained from the different field-based tests and separately by type of reliability. The random effect models for the Biering-Sorensen test showed an average intra-tester and inter-session reliability of *ICC+* = 0.88 (95% CI = 0.80 to 0.92, *I^2^* = 88.9%) obtained from 27 cohorts, an average intra-tester and intra-session reliability of *ICC+* = 0.88 (95% CI = 0.83 to 0.92, *I^2^* = 51.2%) from 12 cohorts, and an average inter-tester reliability of *ICC+* = 0.94 (95% CI = 0.84 to 0.98, *I^2^* = 93.1%) from eight cohorts. The prone isometric chest raise test reported an average intra-tester and inter-session reliability of *ICC+* = 0.95 (95% CI = 0.91 to 0.97, *I^2^* = 70.5%) from five cohorts. Finally, the average intra-tester and inter-session reliability of the dynamic extensor endurance test was *ICC+* = 0.99 (95% CI = 0.88 to 1.0, *I^2^* = 74.5%). Figure 2 displays a summary of the intra-tester and inter-session reliability obtained in the studies that applied the Biering-Sorensen test.

In the different meta-analyses carried out, the effect sizes exhibited a moderate to large heterogeneity (based on the *Q* statistics and the *I^2^* indices), supporting the decision of applying the random-effects model.

#### 3.4.2. Analysis of the Moderator Variables

As only the intra-tester and inter-session reliability for the Biering-Sorensen test presented more than 20 reliability estimates, the analyses of moderator variables were carried out exclusively for this field-based test and type of reliability, in order to examine the influence of qualitative and continuous moderator variables on the reliability coefficients.

Table 3 shows the results of the meta-regression analyses for each continuous moderator variable. None of the continuous moderators exhibited a statistically significant relationship with the intra-tester and the inter-session reliability of the Biering-Sorensen test.

Regarding the qualitative moderator variables, Table 4 shows the ANOVA results. Concerning the description of the field-based tests, significant statistical differences were obtained in position control instruments (*p* < 0.0001) with an explained variance of 69%. Specifically, the studies that used a plumb-line to control position exhibited, on average, greater reliability (*ICC+* = 0.99), followed by the stadiometer (*ICC+* = 0.89), the light sensor (*ICC+* = 0.87), the inclinometer (*ICC+* = 0.83), and the visual control (*ICC+* = 0.81). Similarly, statistically significant differences were observed (*p* = 0.045) between the use of a roman chair (*ICC+* = 0.94) or a test bench (*ICC+* = 0.85), with an explained variance of 32%.

Concerning the characteristics of the study design, the presence of a familiarization session before data collection showed a statistically significant relationship with the reliability estimates (*p* = 0.0005). Specifically, carrying out a familiarization session showed larger average reliability (*ICC+* = 0.96) than when it was not performed (*ICC+* = 0.82), with an explained variance of 34%.

Finally, the sex of the sample approached statistical significance on the average reliability (*p* = 0.05, *R^2^* = 0.27), with the female samples presenting a better reliability (*ICC+* = 0.97) than the males (*ICC+* = 0.88) or males and females together (*ICC+* = 0.83). Similarly, the sample type showed a tendency towards statistical significance (*p* = 0.053, *R^2^* = 0.18). The studies that used children or adolescents as the sample showed, on average, higher reliability (*ICC+* = 0.95) than those that tested adults (*ICC+* = 0.85).

### 3.5. An Explanatory Model

As a further step, a multiple meta-regression was applied to identify the subset of moderator variables that can explain most of the reliability coefficient variability. The predictors included in the model were selected on the basis of the ANOVAs and meta-regression results previously conducted. Thus, out of the five moderator variables that presented a statistically significant relationship with the reliability coefficients (sex, sample type, tool, position control instruments, and familiarization session), a multiple regression model with two predictors was the model that best explained the variability of the reliability coefficients—the type of sample (0, adults vs. 1, children/adolescents) and the familiarization session (0, No vs. 1, Yes). Table 5 presents the results. These two moderator variables reached statistical significance (*Q_R_* (2) = 20.57, *p* < 0.001) with 51% of variance accounted for. As shown in Table 4, both moderator variables exhibited a statistically significant relationship with the reliability coefficients, once the influence of the other moderator was controlled.

### 3.6. Reliability Induction

Out of the 103 empirical studies that reported some reliability coefficients, 75 studies induced reliability (37 by omission and 38 by the report from other studies), which implies a 72.8% of RI. In particular, of these 38 studies that induced reliability by the report from other studies (41 cohorts), 32 induced reliability of the Biering-Sorensen test, 4 the prone isometric chest raise test, 2 the supine bridge test, 1 the prone double straight-leg raise test, and 2 the reliability of the dynamic extensor endurance test (see Appendix A).

## 4. Discussion

The main purpose of the current RG meta-analysis was to estimate both the inter- and intra-tester (intra- and inter-session) reliability of the trunk extension endurance measures obtained through four field-based tests (Biering-Sorensen, prone isometric chest raise, prone double straight-leg raise, and dynamic extensor endurance), as well as to identify those characteristics (qualitative and quantitative moderators) of the studies selected that might have a meaningful influence in the degree of heterogeneity showed by the pooled reliability coefficients (ICC scores). A secondary purpose was to determine the RI practice in studies that used at least one of the four trunk muscle endurance field-based tests investigated in this RG meta-analysis.

The systematic literature review carried out showed that very few studies have analyzed the inter-tester reliability of the trunk extension endurance measures obtained through field-based tests [18,23,24,26,66]. Specifically, most of these studies focused on the analysis of the inter-tester reliability of the trunk extension endurance measure obtained through the Biering-Sorensen test [18,23,24,26,66] (eight cohorts); the number of studies that explored the inter-tester reliability of the endurance measures recorded from the prone isometric chest raise [18] (one cohort) and the prone double straight-leg [18] (one cohort) tests were very limited, and no studies used the dynamic extensor endurance test. In particular, the results of this RG meta-analysis showed appropriate inter-tester reliability values (*ICC+* > 0.80) for the trunk extension endurance measures obtained through the Biering-Sorensen (*ICC+* = 0.94), the prone chest raise (*ICC+* = 0.90), and the prone double straight-leg (*ICC+* = 0.83) tests. However, and due to the scarce evidence available (one cohort with a small sample size (*n* = 30) was found), the inter-tester reliability estimates of the measures obtained through the prone isometric chest raise and prone double straight-leg tests should be considered with a degree of caution.

On the other hand, five studies [18,25,26,43,67] analyzed the intra-tester reliability of the trunk extension endurance measures, using short periods (intra-session, within a day). Most of these studies focused exclusively on the endurance measure obtained through the Biering-Sorensen test, which reported appropriate reliability results (*ICC+* = 0.88). For the rest of the trunk extensor endurance field-based tests, only one study was found that assessed the intra-tester reliability in its intra-session modality for the prone isometric chest raise and the prone double straight-leg tests [18], reporting acceptable reliability results (*ICC+* = 0.90 and 0.86, respectively). However, these intra-tester reliability results for the prone isometric chest raise and the prone double straight-leg tests were not supported by strong evidence (only 1 cohort of 30 participants [18]) and hence should be considered with a degree of caution. There appears to be no study focusing on the intra-session reliability of the measure obtained from the dynamic extensor endurance test.

Contrarily, there was a higher number of studies that analyzed the intra-tester and inter-session reliability of the trunk extension endurance measures obtained through the field-based tests selected (except the endurance measures obtained from the prone double straight-leg test, as no study has analyzed its inter-session reliability). Different authors [70,71] have suggested that the scarce number of intra-session reliability coefficients reported in the literature may be explained by the fact that field-based tests are often used in longitudinal randomized controlled trials that analyze the chronic effects of training interventions rather than in studies that were conducted using short periods to analyze the acute effects of specific interventions. Therefore, the higher number of studies that have analyzed the inter-session reliability in contrast with those that have analyzed the intra-session reliability of the endurance measures seems to be in line with the research designs and interests currently present in the scientific literature. All the trunk extension measures analyzed reported appropriate inter-session reliability scores, with ICC values ranging from *ICC+* = 0.88 (Biering-Sorensen test) to 0.99 (dynamic extensor endurance test).

On the whole, only the trunk extension endurance measure obtained through the Biering-Sorensen test may present sufficient scientific evidence in terms of inter- (*ICC+* = 0.94, based on 8 cohorts) and intra-tester (intra-session (*ICC+* = 0.88, based on 12 cohorts) and inter-session (*ICC+* = 0.88, based on 27 cohorts)) reliability to justify its use for practical and clinical purposes.

The analysis of the quantitative and qualitative factors or moderators that may have had an influence in the degree of heterogeneity found for the pooled reliability coefficients of the trunk extension endurance measures could only be carried out in the measurements obtained through the Biering-Sorensen test and for the inter-session subtype of intra-tester reliability because it was the only one with at least 20 reported reliability estimates. Thus, the results derived from the meta-regressions demonstrated that none of the factors/objects of the study had a statistically significant impact (*p* > 0.05) on the heterogeneity exhibited by the pooled inter-session reliability coefficient of the endurance measure obtained through the Biering-Sorensen test. It should be highlighted that even though a non-statistically significant relationship was found between the moderator “SD test score from reliability sample” and the inter-session reliability coefficient of the Biering-Sorensen test, it explained 10% of the variance. This finding agrees with the psychometric theory that states that the larger the SD of test scores, the greater the reliability that might be obtained [72].

Regarding the qualitative moderator analysis (ANOVAs), the instrument or tool used to control the participant’s position during the test (no instrument (visually controlled) vs. inclinometer vs. stadiometer vs. light sensor vs. plumb-line), the equipment used to perform the test (test bench vs. roman chair) and the presence (or not) of a familiarization session were the moderators that showed statistically significant effects (*p* < 0.05) on the inter-session reliability estimate of the endurance measure obtained through the Biering-Sorensen test.

During the execution of the Biering-Sorensen test, small movements sometimes occur naturally while participants are trying to maintain the required isometric horizontal position. These little movements may make the tester falsely assume a perception of fatigue in the participant who is being tested [23]. To minimize this source of error, it has been recommended that the tester should control the participant’s position during the test through the use of a static visual reference. The results of this RG meta-analysis showed that the use of a plumb-line to control the participant’s position (probably because, and unlike the other tools that only offer visual feedback of the participant’s position to the tester, it also provides kinesthetic feedback in the low back region to the participant who is being tested about the position that has to be maintained during the test) was the instrument that reported the highest inter-session reliability (*ICC+* = 0.99), followed by the stadiometer (*ICC+* = 0.89), light sensor (*ICC+* = 0.87), inclinometer (*ICC+* = 0.83), and visual control (*ICC+* = 0.81).

The tool or equipment used to perform the Biering-Sorensen test also showed significant effects on the inter-session reliability estimate. Those studies that used the roman chair to perform the Biering-Sorensen test reported better inter-session reliability scores (*ICC+* = 0.94) than those using the test bench (*ICC+* = 0.85). However, this difference might be partially attributed to the imbalance distribution existing regarding the number of cohorts selected to carry out the inter-session reliability analysis of both sub-categories of this moderator. Thus, whereas five cohorts were used to calculate the inter-session reliability estimate of the trunk extensor endurance measure obtained through the Biering-Sorensen test using a roman chair, 22 cohorts were used to conduct the same analysis using a test bench.

One of the main sources of error in reliability studies is the presence of learning effects [14]. This effect was identified, as in studies in which participants were asked to carry out a familiarization session for the Biering-Sorensen test before the data collection reported statistically significantly better inter-session reliability coefficients (*ICC+* = 0.96) than those that did not employ a familiarization session (*ICC+* = 0.82). The use of a familiarization session before the assessment of the trunk extensor endurance may help the participant to learn, for example, the appropriate testing position needed to be adopted for the test and to better tolerate the fatigue feelings generated during the Biering-Sorensen test [10,56]. However, quite often it is impossible to carry out a familiarization session in practical (clinic, sports, fitness, and physical education) contexts. Although this circumstance affects the inter-session reliability estimate of the Biering-Sorensen test negatively, the magnitude of this effect is not large enough to consider this endurance measure inappropriate (according to the 0.8 cut-off score) for practical and research purposes.

Other qualitative moderators such as sex (males vs. females vs. males and females) and population of sample (children and adolescents vs. adults) also showed a tendency towards statistical significance. However, and similar to what was found for the moderator tool or equipment, these tendencies may be caused by the large imbalanced distribution shown in the number of cohorts included in each of the sub-categories of these moderators (e.g., males = 9 cohorts vs. females = 2 cohorts vs. males and females = 15 cohorts; adults = 23 cohorts vs. children and adolescents = 4 cohorts). Despite this, the sub-categories of these two moderators showed reliability estimates higher than 0.8, and hence the Biering-Sorensen test may be a suitable field-based test to be conducted in both sexes and youth and adult populations.

Finally, out of 103 studies that reported reliability coefficients, 75 studies induced reliability (72.8%). The number of studies that induced reliability according to our study is in concordance with the high rate of RI that usually characterizes research in general and which is around 70% of the studies conducted in the social and health sciences [73]. These findings reinforce the need to educate researchers that reliability is not an inherent characteristic of the instrument and that it should be analyzed whenever a test is administered.

## 5. Limitations

This RG meta-analysis presents some limitations that should be highlighted. First, the unadvised practice of inducing reliability reduced the number of studies included in this RG meta-analysis. Second, only studies in English or Spanish were included, which may have limited the number of potentially eligible studies. Third, the lack of important data reported by authors reduced the possibility of analyzing their influence as potential moderating variables on the reliability coefficients. Such was the case of the age means and age standard deviation of the samples, as well as the sex distribution, the physical activity level, or the tester’ profession and level of experience conducting the test. To finish, the scarce number of reliability estimates reported for the field tests other rather than the Biering-Sorensen test did not allow to analyze the moderator variables that may explain in part the variability found among the reliability coefficients for those tests.

## 6. Conclusions

The main findings of the current RG meta-analysis report that only the trunk extension endurance measure obtained through the Biering-Sorensen test may present sufficient scientific evidence in terms of inter- (*ICC+* = 0.94, based on 8 cohorts) and intra-tester (intra-session (*ICC+* = 0.88, based on 12 cohorts) and inter-session (*ICC+* = 0.88, based on 27 cohorts)) reliability to justify its use (mainly in male adults) for practical and clinical purposes. Furthermore, this trunk extension endurance measure may be even more reliable when a familiarization session with the Biering-Sorensen testing procedure is previously carried out and a plumb-line is used to control the participant’s position during the test.

None of the rest of the trunk extension endurance measures obtained through the other static (prone isometric chest raise test and prone double straight-leg raise test) and dynamic field-based (dynamic extensor endurance test) tests showed sufficient scientific evidence (in terms of reliability) to promote their use for practical and clinical purposes. Therefore, more reliability studies conducted in different cohorts are needed before the use of these field-based tests can be recommended to assess trunk extensor endurance in both practical and research settings.

Finally, the high RI rate found in this study (72.8%) suggests that researchers should be more aware of the fact that RI is an erroneous practice that should be eradicated because it can cause errors in the estimation of the measures used [74]. Thus, in future research, authors should report the reliability coefficient of the scores for the data being analyzed, even when the focus of their research is not psychometric.

## Figures and Tables

**Figure 1 ijerph-17-03088-f001:**
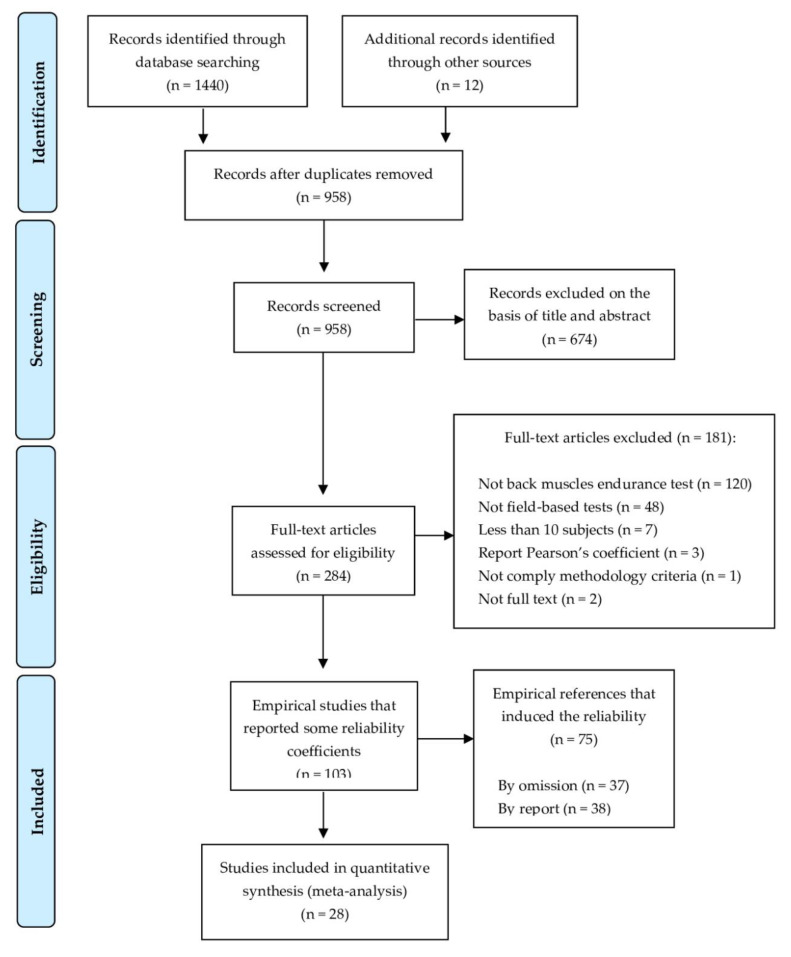
The Preferred Reporting Items for Systematic Reviews and Meta-Analyses (PRISMA) flow diagram of the literature search.

**Figure 2 ijerph-17-03088-f002:**
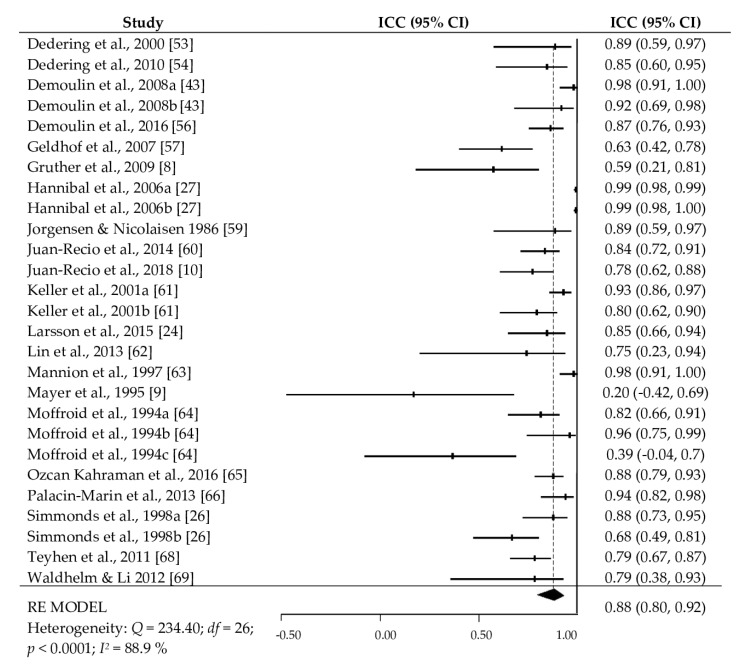
Forest plot of the intra-tester and inter-session reliability obtained in the studies that applied the Biering-Sorensen test.

**Table 1 ijerph-17-03088-t001:** Summary of findings (SoF).

Reliability	Pooled Result ^a^	Overall Rating ^b^	Quality of Evidence ^c^
**Biering-Sorensen Test**			
Inter-tester reliability	ICC = 094 (0.84–0.98) Consistent results Sample size = 215	Sufficient	Moderate (as there are multiple doubtful [18,23,27] and two very good studies [64,69])
Intra-tester (intra-session) reliability	ICC = 0.88 (0.83–0.92) Consistent results Sample size = 258	Sufficient	Low (as all studies are doubtful [18,25,27,44,70])
Intra-tester (inter-session) reliability	ICC = 0.88 (0.80–0.92) Consistent results Sample size = 688	Sufficient	Moderate (as there are multiple doubtful and five very good studies [8,10,61,63,69])
**Prone Isometric Chest Raise Test**			
Inter-tester reliability	ICC = 0.90 (0.80–0.95) Sample size = 30	Indeterminate	Did not pool the results or grade the evidence due to there being one study available [18]
Intra-tester (intra-session) reliability	ICC = 0.90 (0.83–0.94) Sample size = 30	Indeterminate	Did not pool the results or grade the evidence due to there being one study available [18]
Intra-tester (inter-session) reliability	ICC = 0.95 (0.91–0.97) Consistent results Sample size = 236	Sufficient	Low (as all studies are doubtful [11,56,60])
**Prone Double Straight-Leg Test**			
Inter-tester reliability	ICC = 0.83 (0.67–0.93) Sample size = 30	Indeterminate	Did not pool the results or grade the evidence due to there being one study available [18]
Intra-tester (intra-session) reliability	ICC = 0.86 (0.77–0.92) Sample size = 30	Indeterminate	Did not pool the results or grade the evidence due to there being one study available [18]
**Dynamic Extensor Endurance Test**			
Intra-tester (inter-session) reliability	ICC = 0.99 (0.88–1.00) Consistent results Sample size: 82	Sufficient	Low (as there is one inadequate study [65])

^a^ Pooled results obtained from mean reliability analysis and adjusted according to the publication bias analysis. ^b^ Overall rating was graded as sufficient (intraclass correlation coefficient (ICC) > 0.70), insufficient (ICC < 0.70), or indeterminate (either ICC reported by just one study or by none). ^c^ Quality of evidence (high, moderate, low, very low) based on the Grading of Recommendations Assessment, Development and Evaluation (GRADE) approach that uses three factors: (1) risk of bias assessed with the consensus-based standards for the selection of health measurement instruments (COSMIN) risk of bias check-list; (2) inconsistency, solved by pooling the results; and (3) imprecision, the total sample included in the studies. When the total sample size of the pooled studies is below 100, downgrade with one level and with two levels when the total sample size is below 50. ICC: intraclass correlation coefficient.

**Table 2 ijerph-17-03088-t002:** Average reliability, 95% confidence intervals, and heterogeneity statistics for each field-based test and reliability type.

	*k*	*ICC* _+_	*ICC* _L_	*ICC_U_*	*Q*	*df*	*p*	*I* ^2^
**Biering-Sorensen Test:**								
Inter-tester reliability	8	0.94	0.84	0.98	102.37	7	<0.001	93.1%
Intra-tester and intra-session reliability	12	0.88	0.83	0.92	22.69	11	0.02	51.2%
Intra-tester and inter-session reliability	27	0.88	0.80	0.92	234.40	26	<0.001	88.9%
**Prone Isometric Chest Raise test:**								
Inter-tester reliability	1	0.90	0.80	0.95	--	--	--	--
Intra-tester and intra-session reliability	2	0.90	0.83	0.94	0.034	1	0.853	0%
Intra-tester and inter-session reliability	5	0.95	0.91	0.97	13.57	4	0.009	70.5%
**Prone Double Straight-Leg Test:**								
Inter-tester reliability	1	0.83	0.67	0.92	--	--	--	--
Intra-tester and intra-session reliability	2	0.86	0.77	0.92	0.08	9	0.777	0%
**Dynamic Extensor Endurance Test:**								
Intra-tester and inter-session reliability	5	0.99	0.88	1.00	15.73	4	0.003	74.5%

*k*: number of cohorts, *ICC_+_*: mean intraclass correlation coefficient, *ICC_L_* and *ICC_U_*: 95% lower and upper CI for *ICC_+,_ Q*: heterogeneity statistic, *DF*: degrees of freedom for *Q* statistic, *p*: probability level associated to *Q* statistic, *I*^2^: heterogeneity index.

**Table 3 ijerph-17-03088-t003:** Results of the mixed-effects meta-regressions for the continuous moderator variables on the intraclass correlation coefficient (ICC) estimates obtained from intra-tester and inter-session reliability of the Biering-Sorensen test.

Moderator Variable	*k*	*b_j_*	*Q_R_*	*p*	*Q_E_*	*R^2^*
Publication year	27	0.003	0.03	0.855	232.21 *	0
Final sample size	27	−0.003	0.11	0.741	230.26 *	0
Sex (% female)	24	0.002	0.37	0.542	200.78 *	0
Mean age (years)	26	−0.017	1.62	0.203	210.81 *	0.01
SD age	19	−0.009	0.31	0.578	177.94 *	0
% attrition	27	−0.008	1.39	0.238	227.47 *	0
Number of measurements	27	−0.034	0.01	0.933	232.93 *	0
Time interval between measurement	26	−0.006	0.18	0.668	229.87 *	0
Mean test score from total sample	26	0.002	0.98	0.321	210.60 *	0
SD test score from total sample	21	0.009	0.94	0.331	201.42 *	0.02
Mean test score from reliability sample	24	0.001	0.42	0.514	207.27 *	0
SD test score from reliability sample	20	0.012	1.54	0.214	180.06 *	0.10

*k*: number of studies, *b**_j_*: regression coefficient for the predictor variable, *Q_R_*: statistic for testing the statistical significance of the predictor variable, *p*: probability level for the *Q_R_* statistic, *Q_E_*: statistic for testing the model misspecification, *R*^2^: proportion of variance explained by the predictor variable, * *p* < 0.001.

**Table 4 ijerph-17-03088-t004:** Results of the mixed-effects ANOVAs for the qualitative moderator variables on the intraclass correlation coefficient (ICC) estimates obtained from intra-tester and inter-session reliability of the Biering-Sorensen test.

Moderator Variables	*k*	*ICC_+_*	95% CI		ANOVA Results
			*ICC* *_L_*	*ICC_U_*	
Reliability analysis was done with the same sample:					*Q_B_* (1) = 0.92, *p* = 0.338; *R^2^* = 0.01 *Q_W_* (25) = 217.83, *p* < 0.0001
Yes	24	0.88	0.81	0.92	
No	3	0.77	0.26	0.94	
Sex:	9				*Q_B_* (2) = 5.98, *p* = 0.050; *R^2^* = 0.27 *Q_W_* (23) = 156.41, *p* < 0.0001
Males	2	0.88	0.77	0.94	
Females	15	0.97	0.89	0.99	
Males and females		0.83	0.72	0.89	
Sample type:					*Q_B_* (1) = 3.73, *p* = 0.053; *R^2^* = 0.18 *Q_W_* (25) = 185.35, *p* < 0.0001
Children and adolescents	4	0.95	0.86	0.98	
Adults	23	0.85	0.77	0.90	
Target population:					*Q_B_* (1) = 0.24, *p* = 0.625; *R^2^* = 0 *Q_W_* (25) = 231.60, *p* < 0.0001
Asymptomatic	18	0.88	0.80	0.93	
Clinical	9	0.85	0.69	0.93	
Physical activity level:					*Q_B_* (1) = 0.78, *p* = 0.332; *R^2^* = 0.03 *Q_W_* (25) = 214.29, *p* < 0.0001
Sedentary	12	0.84	0.71	0.92	
Recreationally active	15	0.89	0.81	0.94	
Validated modification:					*Q_B_* (1) = 0.24, *p* = 0.626; *R^2^* = 0 *Q_W_* (25) = 222.31, *p* < 0.0001
Yes	17	0.86	0.76	0.92	
No	10	0.89	0.78	0.95	
Tool:					*Q_B_* (1) = 4.03, *p* = 0.045; *R^2^* = 0.32 *Q_W_* (25) = 158.11, *p* < 0.0001
Test bench	22	0.85	0.77	0.90	
Roman chair	5	0.94	0.86	0.98	
Hands position:					*Q_B_* (2) = 0.37, *p* = 0.828; *R^2^* = 0 *Q_W_* (24) = 217.66, *p* < 0.0001
Crossed on the chest	22	0.88	0.80	0.93	
Along the body	3	0.87	0.55	0.97	
At the level of the ears	2	0.80	0.23	0.96	
Part of the body on the edge:					*Q_B_* (3) = 1.26, *p* = 0.738 4; *R^2^* = 0 *Q_W_* (23) = 212.02, *p* < 0.0001
Not reported	12	0.90	0.80	0.95	
ASIS	6	0.88	0.68	0.95	
Upper border of the iliac crest	6	0.85	0.62	0.95	
Pubis	3	0.77	0.28	0.94	
Test duration:					*Q_B_* (1) = 1.24, *p* = 0.264; *R^2^* = 0.01 *Q_W_* (25) = 212.46, *p* < 0.0001
Until exhaustion	25	0.88	0.82	0.93	
Until 240 s	2	0.72	0.08	0.94	
Position control instruments:					*Q_B_* (4) = 34.49, *p* < 0.0001; *R^2^* = 0.69 *Q_W_* (22) = 72.29, *p* < 0.0001
Visual	14	0.81	0.73	0.87	
Inclinometer	5	0.83	0.67	0.91	
Stadiometer	4	0.89	0.78	0.95	
Light sensor	2	0.87	0.59	0.96	
Plumb-line	2	0.99	0.98	0.99	
Familiarization session:					*Q_B_* (1) = 12.09, *p* = 0.0005; *R^2^* = 0.34 *Q_W_* (25) = 151.63, *p* < 0.0001
Yes	6	0.96	0.92	0.98	
No	20	0.82	0.73	0.88	
Test conditions:					*Q_B_* (1) = 0.27, *p* = 0.605; *R^2^* = 0 *Q_W_* (25) = 222.08, *p* < 0.0001
Similar conditions	18	0.88	0.80	0.93	
Unclear conditions	9	0.85	0.68	0.93	
The profession of tester:					*Q_B_* (2) = 1.88, *p* = 0.389; *R^2^* = 0.02 *Q_W_* (23) = 205.35, *p* < 0.0001
Sports sciences	6	0.92	0.80	0.97	
Physical therapy	17	0.86	0.76	0.92	
Medicine	3	0.77	0.30	0.94	
Continent:					*Q_B_* (1) = 0.0004, *p* = 0.984; *R^2^* = 0 *Q_W_* (24) = 226.96, *p* < 0.0001
Europe	16	0.88	0.78	0.93	
America	10	0.88	0.74	0.94	
Study objective:					*Q_B_* (1) = 0.02, *p* = 0.877; *R^2^* = 0 *Q_W_* (25) = 231.27, *p* < 0.0001
Psychometric	20	0.88	0.79	0.93	
Not psychometric	7	0.86	0.67	0.95	
Conflict of interest:					*Q_B_* (1) = 0.02, *p* = 0.896; *R^2^* = 0 *Q_W_* (25) = 232.67, *p* < 0.0001
No conflict	8	0.87	0.71	0.94	
Not reported	19	0.88	0.79	0.93	
COSMIN Risk of Bias check-list:					*Q_B_* (3) = 0.81, *p* = 0.848; *R^2^* = 0 *Q_W_* (23) = 223.47, *p* < 0.0001
Very good	6	0.85	0.62	0.94	
Adequate	2	0.80	0.22	0.96	
Doubtful	17	0.85	0.79	0.93	
Inadequate	2	0.92	0.56	0.99	

*k*: number of cohorts, ASIS: anterosuperior iliac spine, ANOVA: analysis of variance, *ICC_+_*: mean reliability coefficient, *ICC**_L_* and ICC*_U_*: lower and upper 95% confidence limits for *ICC_+_*, *Q_B_*: statistic for testing the statistical significance of the predictor variable, *Q_W_*: statistic for testing the model misspecification, *R^2^*: proportion of variance explained by the predictor variable.

**Table 5 ijerph-17-03088-t005:** Results of the multiple meta-regression model assuming a mixed-effects model.

Source	*b_j_*	*SE*	*Z*	*p*
Intercept	1.09	0.11	9.89	<0.001
Sample type	0.65	0.27	2.39	0.017
Familiarization session	0.68	0.24	2.88	0.004
Full model:	*Q_R_* (2) = 20.57, *R*^2^ = 0.51 *Q_E_* (24) = 113.67, *p* < 0.001			

*b_j_*: unstandardized regression coefficient, SE: standard error of *b_j_*, *Z*: statistic for testing the statistical significance of each moderator variable, *Q_R_*: statistic for testing the global significance of the model, *Q_E_*: statistic for testing the model misspecification, *R*^2^: proportion of variance accounted for by the full model.

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
