# Peer review of "A Meta-Analysis of the Reliability of Four Field-Based Trunk Extension Endurance Tests"

_ijerph, 2020, doi:10.3390/ijerph17093088_

Round 1

Reviewer 1 Report

No further comments.

Reviewer 2 Report

The revisions made by the authors are sufficient.

Reviewer 3 Report

After a rapid check in according to the international Meta Analysis guidelines I suggest to accept this submission.

This manuscript is a resubmission of an earlier submission. The following is a list of the peer review reports and author responses from that submission.

Round 1

Reviewer 1 Report

The article intitled “A meta-analysis of the reliability of four field-based trunk extension endurance tests” aimed to conduct a meta-analysis and be able to show and help practitioners and researcher to choose the best field-based test for trunk extension.

Is article is well written and the authors were very careful to provide all information within the article and in supplementary tables. My only concern is regarding the conclusion. The first paragraph of the conclusion really looks like I’m reading the first paragraph of the discussion. The authors are just showing their main results and starts to explain the results but in the conclusion section the writer expects the reader already know this. Thus, the conclusion should just focus on the extrapolation of the results, focusing on practical applicability of the results, or maybe what should future studies should focus.

Reviewer 2 Report

Lines 22 – 26: The starting sentence of the abstract needs to be re-written. The opening sentence is very long-winded and needs to be more concise.

Introduction

Line 44- 48: Same as above.

Lines 50-53: The writing needs to be a lot more concise throughout the entire article. For example, the following lines could be written like…”Although several sophisticated laboratory-based tests have been developed to quantify trunk extensor muscle endurance (i.e.: force platforms, isokinetic dynamometers) [7–9], field-based tests seem to be the most popular tests as they are easier to administer and more cost effective (Reference).

General comment: Each test that is mentioned needs to be better explained for readers who do not understand each field test. Most of the introduction is focused on the reliability testing, yet more emphasis is required on the tests themselves.

Line 127-131 – Mention the four tests that were considered. A lot of explanation is required to justify why only these back extension exercises were considered and not others.

Line 139 – So studies that used other back extensor endurance tests were included… This seems confusing, why limit introduction to only four exercises and mention them as a set criteria for the meta-analysis?

Table 1- How was each tests reliability graded? This needs to be included or made a lot clearer.

Line 406-409 – Should this be included in the methodology section as opposed to the discussion?

Conclusions – It is suggested in the conclusion that only the Biering-Sorensen test can be considered reliable. No discussion was made about other tests with no suggestion that the poor sample size of studies influenced the meta-analyses. In certain circumstances, the ICC was stronger for these other tests when compared to the Biering-Sorensen test, however, this is not mentioned.

Reviewer 3 Report

This paper addresses an important problem regarding the reliability of field testing and examines the reliability of field testing for trunk extensor endurance. The manuscript is generally well-written, has good logical flow, and contains the necessary information in each section.

I have no major concerns with the design or interpretation of the study but do have some minor comments that I feel need to be addressed to make this manuscript publication-ready.

Minor comments:

line 169: "which" should be "that" as this is a restrictive clause

In Figure 2, it is difficult to read the x-axis values for the ICC. Please increase the font size.

There is a high degree of wordiness in the manuscript. Due to the long length and the highly technical analysis, I strongly recommend a careful revision to reduce wordiness. In particular, the first sentence of the discussion is extremely wordy.
